# Repulsive Force for Micro- and Nano-Non-Contact Manipulation

Amélie Cot [1,2], Patrick Rougeot [1], Sophie Lakard [2], Michaël Gauthier [1,]*[ID] and Jérôme Dejeu [1,3,]*[ID]

1    FEMTO-ST Institute, SUPMICROTECH, CNRS, Université Bourgogne Franche-Comté, 25000 Besançon, France
2    UTINAM Institute, Université Franche-Comté, 16 Route de Gray, 25030 Besançon, France
3    Department of Molecular Chemistry (DCM), CNRS, UMR 5250, Université Grenoble-Alpes, 38000 Grenoble, France
*    Correspondence: michael.gauthier@femto-st.fr (M.G.); jerome.dejeu@univ-grenoble-alpes.fr (J.D.)

**Featured Application: This functionalization is able to provide solutions in order to improve robotic nano- or micromanipulation.**

**Abstract:** Non-contact positioning of micro-objects using electric fields has been widely explored, based on several physical principles such as electrophoresis, dielectrophoresis (DEP) or optical dielectrophoresis (ODEP), in which the actuation force is induced by an electric charge or an electric dipole placed in an electric field. In this paper, we introduce a new way to control charges in non-contact positioning of micro-objects using chemical functionalization (3-aminopropyl) triethoxysilane—APTES) able to localize charges on a substrate and/or on a micro-object. We demonstrate that this functionalization in a liquid with a low ionic strength is able to concentrate a significant amount of electric charges on surfaces generating an electric field over a long distance (about 10 microns), also called a large exclusion zone (EZ). A model is proposed and validated with electrostatic force measurements between substrate and microparticles (diameter up to 40 µm). We demonstrate that the magnitude of the force and the force range decrease rapidly when the ionic strength of the medium increases. Based on the proposed model, we show that this new way to localize charges on micro-objects may be used for non-contact positioning.

**Keywords:** APTES grafting; exclusion zone (EZ); force modeling; non-contact-manipulation; PANI electropolymerization; repulsive force; surface functionalization

## 1. Introduction

Non-contact micro-nanomanipulation or micro-sorting consists the manipulation or sorting of microparticles using forces generated via a long range physical field (magnetic field, electrostatic field, acoustic field). The behavior and the design of the devices are significantly modified by the well-known scale effects [1]. When the scale reduces, the physical effects' magnitude is drastically modified: when the lengths are divided by 10, the volume effects (e.g., weight mass) are divided by $10^3$, and the surface forces (e.g., van der Waals force) are divided only by $10^2$. Therefore, the effect of gravity thus decreases more rapidly than the effect of surface forces during miniaturization [2]. Hence, on the microscale, the surface forces are predominant compared to the weight and the objects which tend to stick to the surfaces (e.g., adhesion [3]). Therefore, the manipulation (movement and position) of a micro-object is usually performed without touching it, but instead by using the proposed "non-contact manipulation" to avoid adhesion disturbance [4]. These manipulation methods are usually propelled by electrostatic [5], laser-induced thermal gradient [6], optical trapping [7,8], magnetic [8,9], or thermocapillary [10] forces. This article presents a new way to perform micromanipulation using electrostatic forces. In this field, the electric field is usually generated by (micro)electrodes placed in a liquid in order to induce dielectrophoresis (DEP) [11], optically induced dielectrophoresis (ODEP) [12],

electro-rotation [13], or electrophoresis [14]. Indeed, applying a voltage on electrodes induces charges generating an electric field up to the electrodes. The microparticles located in the electric field experience a force of several tens of micrometers away from the electrodes depending on the applied electric field generated by the charges of the electrodes, the particle's size, and electrical properties. This general principle is usually used to sort particles [12,15].

The objective of this paper is to propose a new way to control non-contact manipulations based on electric fields. Indeed, we propose to generate electrophoretic force using electric charges based on chemical principles in spite of generating electric charges generated with an external voltage. Concretely, a local chemical functionalization (typically amine groups) enables to locally concentrate electric charges generating an electric field up to the substrate in the liquid [16]. In order to be usable, the electric field generated by chemical functions has to be able to induce significant electrostatic forces on microparticles on a long range (typically greater than one micron).

We demonstrate below the highly impacted interaction distance by the scale effect, showing that nanoparticles and microparticles have significant different behaviors.

In nanoscale, the repulsive forces between colloids and a flat surface were already measured by different researchers in liquid media. The measures of repulsion forces enable to identify the hydration force [17] or electrostatic repulsions in inorganic solvents with different types of spheres glued on atomic force microscope (AFM) tips: alumina or silica [18], silicon nitrite [19], and gold [20] tips. More recently, some repulsive electrostatic forces have been measured in water [21]. In all these cases, the size of the sphere diameter is less than 5 $\mu$m and the interaction distance is near 20 nm.

Our paper focuses on larger microspheres whose diameters are higher than several micrometers. We demonstrate below the possibility to generate significant electrostatic forces on these microparticles over a long range (around tenth micron).

Section 2 introduces the materials and methods used in the technical parts. Section 3 reports some experimental force measurement illustrating the long-range interaction forces. Section 4 focuses on a model of the force whose comparison with experiments is described in Section 5. The simulation of potential uses of these long range forces in non-contact manipulation of microparticles is introduced in Section 6 and the general results are discussed in Section 7 before the conclusion.

## 2. Materials and Methods

### 2.1. Salt

Lithium percholorate (LiClO$_4$), sodium tetrafluoroborate (NaBF$_4$), sodium nitrate (NaNO$_3$) and p-toluenesulfonic acid (PTS) came from Sigma-Aldrich (Saint-Quentin-Fallavier, France).

### 2.2. Media Preparation

AFM measurements were performed in water at pH 2 prepared before each day before the series of measurements. At the end of the experiments, the pH was controlled to validate the measurements performed. The pH of the solution was measured with a pH-meter (Sartorius, PT-10) and an electrode (Sartorius, PY-P22) and adjusted at pH 2 by adding hydrochloric acid 1 M just before the measurement to protonate all the amine functions. For all experiments, ionic strength was about 10–3 M except for experiments concerning ionic strength's influence (controlled by NaCl addition).

### 2.3. Surface Functionalization

Before being functionalized, silicon wafers (purchased from Tracit) were cleaned by immersion in a piranha solution (2 parts H$_2$SO$_4$ and 1 part H$_2$O$_2$) for 25 min at 70 °C. Then, wafers were rinsed in Milli-Q water and ethanol before functionalization.

### 2.4. Silanization

Solutions were freshly prepared by the direct dissolution of silanes (3-aminopropyl) triethoxysilane, APTES) in ethanol. The final silane concentration was 1%. The surfaces were functionalized by immersion in solutions for one night at room temperature. In the silane solution, the molecules were grafted on the substrate (through covalent bonds). The excess of ungrafted silanes was removed by ultrasonication for 2 min in ethanol. The mechanism of SAM formation during the silanization process was already described by Wasserman et al. [22]. The mechanism of self-assembled monolayer formation during the silanization process took place in four steps [22,23]. The first step was physisorption, in which the silane molecules became physisorbed at the hydrated silicon surface. In the second step, the silane head-groups arrived close to the substrate hydrolyse, in the presence of the adsorbed water layer on the surface, into highly polar trihydroxysilane $Si(OH)_3$ for triethoxysilane $Si(OEt)_3$ (APTES). These polar groups, $(Si(OH)_3)$, formed covalent bonds with the hydroxyl groups on the $SiO_2$ surface (third step); subsequently, a condensation reaction (release of water molecules) occurred between the silanol functions of neighbor molecules. Self-assembly was driven by lipophilic interactions between the linear alkane. During the initial period, only a few molecules adsorbed (by steps 1–3) on the surface and the monolayer definitely resulted in a disordered (or liquid) state. However, at longer times, surface coverage eventually reached the point where a well-ordered and compact (or crystalline) monolayer was obtained (step 4) by the condensation reaction between the APTES molecules.

The grafting was controlled by contact angle measurements. The contact angle before functionalization was inferior to 10° and increased from 60° to 80° after APTES grafting. These values were concordant with previous experiments [24].

### 2.5. Electrochemical Deposition of Thin Films

The pyrrole and 3-aminopropyltriethoxysilan were from Acros organics (99% pure) and were distilled under reduced pressure before use. Lithium perchlorate was from Sigma Aldrich and used as electrolytic salt. Electrolytes were composed of 0.1M pyrrole in an aqueous solution of 0.1 M $LiClO_4$ or other salts (PTS, $NaBF_4$, $NaNO_3$). Electrochemical experiments were performed with a PGZ 100 potentiostat (Tacussel-Radiometer Analytical SA-France) controlled by the VoltaMaster 4 software. A standard three-electrode system was relied onto the potentiostat and composed by a saturated calomel electrode (SCE) as the reference electrode, a platinum sheet as the counter-electrode, and a working electrode which was a silicon substrate previously covered by sprayed chrome and gold to enhance its conductivity. The working electrode was cleaned for one hour using a UV-ozone treatment (Bioforce UV/Ozone Procleaner) before the electrochemical deposition process. All electrochemical experiments were carried out at room temperature (293 K). Firstly, cyclic voltammetry technique was used in order to define the potential of polymerization of pyrrole. Then, potentiostatic chronoamperommetries were performed to obtain thin films of polypyrrole. A previous work by Patois et al. [25] showed that polypyrrole could be deposited from +0.7 V/SCE with an oxidation peak appearing at +1.0 V/SCE in the cyclic voltammogram, so we decided to work à +0.7V/SCE. The films obtained were previously characterized by SEM and AFM [26].

### 2.6. Force Measurements

The force measurement experiments were performed with an AFM tip on which a functionalized borosilicate sphere was glued according to the procedure previously described [27]. In order to characterize surface functionalization, a Smena S7 atomic force microscope (AFM) from NTMDT was used. The silicon rectangular AFM cantilever (from Novascan Technologies) had a stiffness of 0.3 N/m. The cantilever was fixed while the substrate moved vertically. Most of the AFM force measurements were made with the tip whose diameter is several tens of nanometers. In order to evaluate the interaction between a micrometer-scaled robot and a substrate, the interaction between a microsphere and a

substrate was considered. Consequently, a borosilicate sphere (from 1 to 40 μm of diameter) was glued onto the cantilever. Force-distance curves were obtained by the measurement's exploitation of the AFM cantilever's deformation measurement from the AFM cantilever with a laser beam and a sensitive four-quadrant photodiode. The measurements were performed at the driving speed of 200 nm/s, to stave off the influence of the hydrodynamic drag forces, in 10 different points minimum, with different surfaces and borosilicate sphere (at least five) for all the conditions tested (ionic strength and borosilicate diameter). All measurements were conducted in a liquid medium using the pH 2 to protonate the chemical groups of interest.

## 3. Results

Substrate and spheres of several sizes from 1 to 40 μm diameters have been functionalized by grafting (3-aminopropyl) triethoxysilane (APTES). The forces between the two surfaces were measured in different points on a sample and on different APTES-modified substrates and AFM tips at pH 2. The surface was kept in the solution for 2 min before starting measurements in order to stabilize the system. Then, the adhesion force was measured with an AFM in which a sphere was glued on the tipless cantilever extremity.

### 3.1. Influence of the Sphere Size on the Repulsive Force Measured

The force distance measurements obtained for APTES-modified surface, with borosilicate spheres of different diameters (between 1 and 40 μm) are presented in Figure 1A.

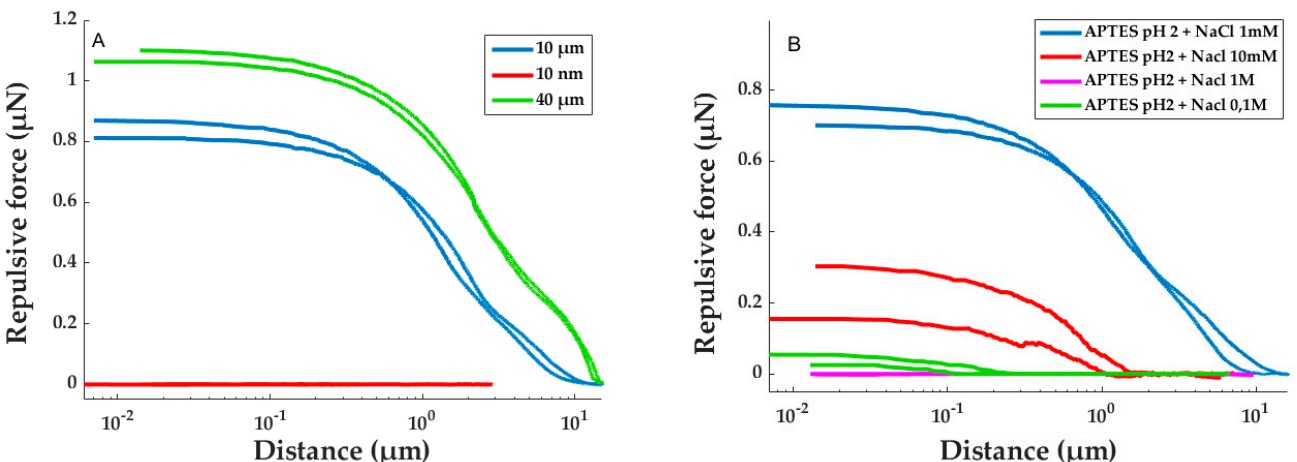

**Figure 1.** AFM force–distance curves at pH 2 between two APTES surfaces (one flat surface and one spherical, glued on the tipless cantilever): (**A**) influence of the functionalized borosilicate sphere: 10 nm (red), 10 μm (blue), and 40 μm (green); (**B**) influence of the ionic strength for a 10 μm borosilicate sphere diameter glued on the tipless: no salt added (blue), 10 mM (red), 0.1 M (green), and 1 M (pink). All the surfaces were functionalized by APTES.

In Figure 1A, a cantilever deformation was observed on a long distance (typically several micrometers) when the sphere was approaching the surface. This distance increased with the size of the sphere from a few nanometers for a 10 nm sphere to 12 μm for a 40 μm sphere. The variation of the repulsive force was similar to the one observed during previous experiments [28]. The size of the sphere glued on the tipless extremity influenced the repulsive force. Indeed, for a lower sphere diameter (10 nm), no significant repulsion was measured, but when the sphere diameter was increased, the repulsive force appeared and it increased with the sphere diameter (Figure 1A). It reached around 800 nN and 1.1 μN for 10 μm and 40 μm spheres, respectively. The repulsion can be explained by the electrostatic repulsion of the positive charges, at pH 2, of the amine grafted on the tip and on the surface.

### 3.2. Influence of the Ionic Strength

To confirm the origin of the repulsion (electrostatic charge), the ionic strength was modified by adding NaCl salt ranging from 10 mM to 1 M in the measuring medium. Then, the force–distance curve was recorded with a 10 μm sphere (Figure 1B). When the medium did not contain any salt, a repulsive force of 700 nN was obtained. The introduction of a small amount of NaCl, to achieve a concentration of 10 mM, resulted in a 3.5-times reduction of the repulsive force (200 nN). By increasing this concentration to 0.1 M, the force did not exceed 50 nN, and finally, for a concentration of 1 M, it became null. Each force curve presented an hysteresis behavior inducing a measure incertitude. However, the incertitude was still below the observed force reduction induced by the increase of the ionic strength. Hence, we can conclude that the origin of the repulsion force is electrostatic.

This conclusion was concordant with the literature where it has been established that the formation of repulsion on a long distance could be explained by a consequent generation of electric field [16,28–31]. However, the Williams' review concluded that a more complete understanding of the mechanisms behind EZ phenomena will assist in understanding their possible roles in biology, as well as their possible engineering applications, such as in microfluidics and filtration [32]. In this context, we decided to model the experimental force based on the electrical layer theory developed by Gouy, Chapman, Stern, and Grahame [33,34].

## 4. Model Development

Usually, force measurements are conducted between a sphere and a planar substrate. A coarse model of the interaction between a charged micro-object and a charged surface has been proposed previously examined [28] based on major assumptions: (i) the object is a sphere; (ii) charges on the object are localized in the center of the sphere; and (iii) the surface is an infinite plane. In this paper, we propose a more precise numerical model able to predict the interaction forces on objects regardless of the shape.

### 4.1. General Case

The presence of a charged surface in an ionic solution induces a specific modification of the medium. If the surface is positively charged, a digressive layer of anions appears around the contact with the surface until the return to the electric equilibrium in the bulk solution. The modeling of object–surface interaction is based on the electrical layer theory developed by Gouy, Chapman, Stern, and Grahame [33,34]. An electrical layer's formation, namely the formation of a compact layer of charged ions opposite to that of the surface, has been modeled by representing the surface as a set of electric dipoles (see Figure 2).

Each elementary dipole on the surface is represented by two electrical charges $+d_{Q_d}$ and $-d_{Q_d}$, separated by a distance $\kappa$. These parameters can be determined for each experimental condition depending on the ionic strength (see Equation (S1) distance). Each elementary dipole induces an electric field up to the substrate whose component $dE_z$ along the vertical axis $z$ is defined by (projection of Coulomb law on $z$ axis):

$$dE_z = \frac{1}{4\pi\varepsilon_0\varepsilon_r}\left(\frac{h_w - \kappa}{MP^3} - \frac{h_w}{MN^3}\right)d_{Q_d}, \tag{1}$$

Each charged particle located in this electric field experiences an electrostatic force in a very similar principle as electrophoresis. Considering an elementary charge $d_{Q_w}$ placed at a point $M$ on an object (Figure 2), the vertical force applied by the elementary dipole on the elementary charge is directly obtained from Equation (1):

$$dF_{elec} = d_{Q_w}dE_z = \frac{1}{4\pi\varepsilon_0\varepsilon_r}\left(\frac{h_w - \kappa}{MP^3} - \frac{h_w}{MN^3}\right)d_{Q_d}d_{Q_w}, \tag{2}$$

Considering that both the substrate and the object have a uniform charge density (expressed in Coulomb.m$^{-2}$), respectively, noted $\Gamma_d$ and $\Gamma_w$, the total vertical force $F_{elec}$ can

be written as the integral of the elementary force on the substrate surface $S_d$ of the plane and the surface of the object $S_w$:

$$F_{elec} = \frac{\Gamma_d \Gamma_w}{4\pi\varepsilon_0\varepsilon_r} \iint\limits_{S_d} \iint\limits_{S_w} \left( \frac{h_w - \kappa}{MP^3} - \frac{h_w}{MN^3} \right) dS_w dS_d \tag{3}$$

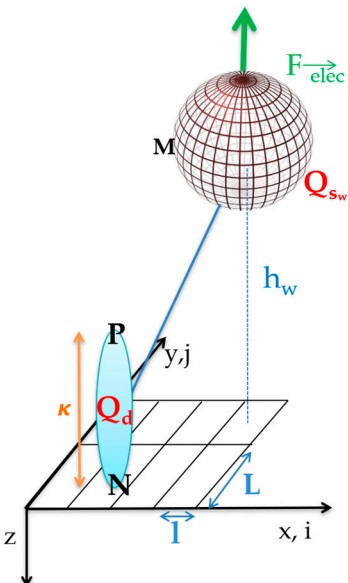

**Figure 2.** Geometric modeling between charged micro-object and surface. The relative scale was not respected: distance $\kappa$ was weaker than the diameter of the sphere. QSw and Qd the charge of the sphere and of the dipole respectively, $\kappa$ the length of the dipole.

### 4.2. Sphere-Plan Case Modeling

In order to compare our model with experimental results, a sphere-plane numerical modeling has been developed. Both the surfaces of the microsphere and of the substrate have been sampled in order to numerically calculate the integral in Equation (3) (see Figure S1). The total force $F_{elec}$ along *z-axis* applied by the substrate on the total sphere is given by:

$$F_{elec} = \frac{\Gamma_d \Gamma_w}{4\pi\varepsilon_0\varepsilon_r} \sum_w \sum_{ij} \left( \frac{h_w - \kappa}{MP(i,j)^3} - \frac{h_w}{MN(i,j)^3} \right) \delta S_d . \delta S_{w\,elec} \tag{4}$$

where $MP$, (respectively, $MN$) represents the distance between the sphere and the top, (respectively, the bottom) of the dipole (Figure 2), which are defined by:

$$MP(i,j) = \sqrt{(h_w - \kappa)^2 + (l.i)^2 + (L.j)^2}, \tag{5}$$

$$MN(i,j) = \sqrt{h_w{}^2 + (l.i)^2 + (L.j)^2}, \tag{6}$$

where $l$ and $L$ are the distances of the sampling of the substrate surface along *x*-axis (*vector i*) and *y*-axis (*vector j*), respectively, and $w$ is the number of samples considered on the object surface ($w = 5000$ in the next sections).

### 4.3. Influence of the Parameters

The dipole thickness, $\kappa$, depends on the ionic strength and also the electrical theory layer. Indeed, the triple electrical layer (Figure S2) is the most complex theory since it takes into account all ions present in the solution in the vicinity of the charged surface. It was modeled as a set of dipoles with different lengths contrary to a simplified model,

called the double electrical layer (Figure S3) where all the dipoles are similar. The major difference between these two models is that the triple layer takes into account the specific adsorption of ions on the surface, which creates a division in the compact layer. We sought to determine whether it was possible to avoid calculating the exponential decay's calculation of the electric potential in the compact layer. The difference between these two theories leads to Figure 3A. The variation of this force $F_{elec}$ versus the distance $z$ is presented, for the two theories. For a numerical application, different sizes of dipoles with a distribution based on the Debye–Hückel approximation were taken into account: a charge density is fixed at 1 charge.nm$^{-2}$ for the silicon substrate $\Gamma_d$ and the borosilicate sphere $\Gamma_w$.

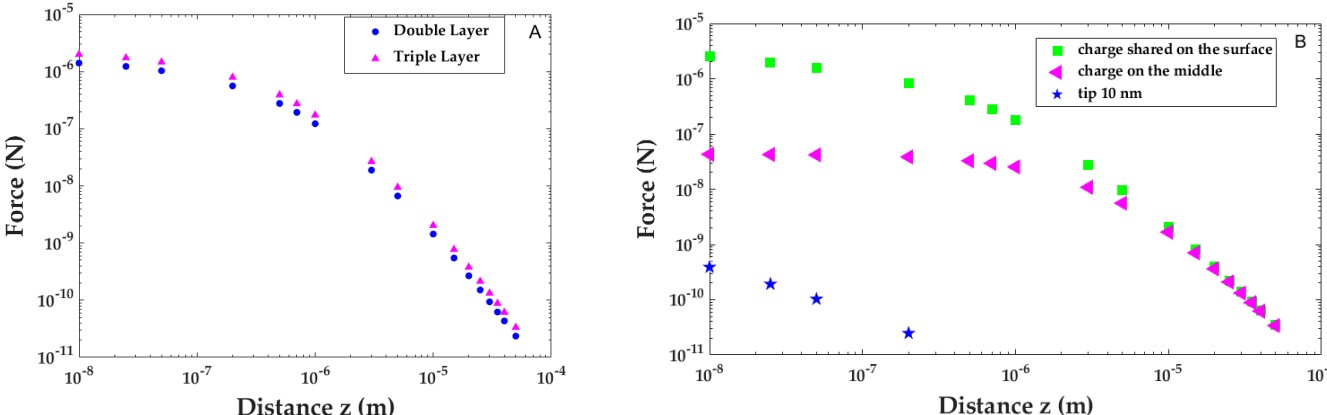

**Figure 3.** (**A**) Modeling of the interaction between a surface covered by dipoles and a charged sphere of 10 μm diameter by considering the simplified double layer (blue sphere) or triple layer model (pink triangles); and (**B**) modeling of the impact of the charge's discretization on sphere surface. Repulsive force obtained between a dipole surface and a 10 μm charged sphere with the charge fairly shared on the surface (green squares), centered at the middle of the sphere as a point charge: the same charge as previous, but located in the sphere center (pink triangles) or 10 nm charged tip (blue stars).

Differences between the two models have a low impact on the considered force and interaction distances. Both models vary only by a factor of less than 2 and the difference is constant on the considered scale. The simplest model (doublelayer model) will be used for the following simulations. The results obtained with both models highlight a long interaction distance as the interaction force at 1 μm distance is only 10 times lower than the interaction force at the contact.

The double sum expressed in Equation (4) was complex to compute due to a long-time calculation. In order to reduce this computational time, we also proposed a simplified model considering that the charges of the objects are located in their centers of gravity (Figure 3B, pink triangles). This elementary charge has the same value as the sum of initial distributed charges on the entire sphere surface. For the 10 μm sphere, full (green squares) and simplified models (pink triangles) are similar until a distance of 10 μm between the sphere and the substrate, then the simplified model underestimates the force. Indeed, in the full model, charges that are at the bottom of the sphere induce a bigger force than those located at the equator. Therefore, it is thus possible to simplify the simulations when the distances of interaction are high enough, at least at the diameter of the considered sphere. Furthermore, it is necessary to use the complete model for the weak interaction distances.

As most of the AFM force measurements are performed with AFM tips having an apparent radius of several nanometers (10 to 100 nm typically) [20,22,35], we propose to compare our simulated results on microspheres with a sphere of 10 nm, modeling the interaction with an AFM tip (Figure 3B, blue stars). The comparison between microsphere and nanosphere shows that the force is lower on nanosphere even for the same charge density. The most important difference is the interaction distance which is significantly lower on nanospheres (few tens of nm) than the distance on microspheres (few μm). This

result is coherent with the current literature, assuming that the electrostatic interaction between a substrate and a nanosphere cannot exceed few tens of nm.

## 5. Model and Experimental Measurements Comparison

The experimental adhesion forces measured were compared with the ones obtained with our model. In this model, the charges of the surface and of the sphere were necessary. Hence, the zeta potential of the borosilicate sphere with and without APTES functionalization was measured with a nanosizer (Malvern). At pH 2, the zeta potential was 34.6 mV and 4.3 mV, corresponding to 0.29 $\mu C/cm^2$ and 0.03 $\mu C/cm^2$, with and without APTES functionalization, respectively. Hence, the APTES-modified sphere charge density was 0.02 charge/$nm^2$. In the model, we decide to also fix the charge density on the surface at the same value, 0.02 charge/$nm^2$, for all the modeling. Using these parameters, the experimental and modeled data simulated using Matlab Simulink software R2020b was compared.

### 5.1. Influence of the Sphere Size on the Repulsive Force Measured

The influence of the sphere size on the interaction distance and the maximum repulsive force predicted by the model (Figure S5) and subsequently measured are presented in Table 1. The factor $\kappa$ (calculated from Equation (S1)) for $10^{-3}$M of ionic strength and $r_{Sw}$ are fixed (and not fitting) depending on the experimental conditions.

**Table 1.** Comparison of the repulsive force and distance measured or modeled between the silica surface and different diameter sizes of borosilicate sphere, both of which are functionalized by APTES. The data are collected during three experimental campaigns. For every campaign, new bead and substrate are considered, and force is measured in 10 different locations on a substrate.

| Sphere Diameter (µm) | Experimental | | Predicted | |
|---|---|---|---|---|
| | Interaction Distance (µm) | Interaction Force (µN) | Interaction Distance (µm) | Interaction Force (µN) |
| 0.01 | $0.004 \pm 0.007$ | $0.02 \pm 0.01$ | N.d. | $1.10^{-4}$ |
| 1 | $0.8 \pm 0.2$ | $0.07 \pm 0.01$ | $0.5 \pm 0.05$ | $0.03 \pm 0.005$ |
| 5 | $3.3 \pm 0.5$ | $0.6 \pm 0.2$ | $3 \pm 0.3$ | $0.34 \pm 0.05$ |
| 10 | $5.7 \pm 0.7$ | $0.7 \pm 0.15$ | $5 \pm 0.5$ | $0.64 \pm 0.1$ |
| 20 | $8.2 \pm 0.6$ | $0.9 \pm 0.25$ | $7.5 \pm 0.75$ | $1.2 \pm 0.2$ |
| 40 | $12.7 \pm 0.29$ | $1.1 \pm 0.12$ | $12.5 \pm 1.2$ | $2 \pm 0.4$ |

N.d. the distance was too weak in order to be estimated.

The experimental data (force values and distance of repulsion) are consistent with the results of the modeling. Indeed, in all cases, the same order of magnitude was obtained. The variation of the difference between the expected and the experimental repulsive force could be explained by a small variation of the charge density. Indeed, each sphere was functionalized with an individual APTES solution, and so, the grafting percentage could be slightly changed between each sphere diameter. The charge density of each sphere was not measured individually. For the predicted value, we take an average value of the density charge measured on a 10 µm radius borosilicate sphere functionalized by APTES. It should be noted that the forces obtained are substantially greater than the weight of the objects considered (11 pN for the 10 µm diameter borosilicate sphere).

### 5.2. Influence of the Ionic Strength

As the behavior of these objects is essentially governed by these electrostatic forces, the impact of the ionic force on both the experimental data and the model has been tested and is reported in Table 2 (Figure S6). For that, the distance $\kappa$ was calculated from Equation (S1) for each experimental condition and $r_{Sw}$ is fixed at 5 µm.

**Table 2.** Comparison of the repulsive force and distance measured or modeled between the silica surface and a 10 μm borosilicate sphere, both of which are functionalized by APTES for different ionic strengths controlled by the addition of NaCl. The data are collected during three experimental campaigns. For every campaign, new bead and substrate are considered, and force is measured in 10 different locations on a substrate.

| | Experimental | | Modeling | |
|---|---|---|---|---|
| Ionic Strength: NaCl (M) | Interaction Force (μN) | Interaction Distance (μm) | Interaction Force (μN) | Interaction Distance (μm) |
| 0 | 0.7 ± 0.15 | 5.7 ± 0.7 | 0.6 ± 0.1 | 5 ± 0.5 |
| 0.01 | 0.2 ± 0.1 | 0.5 ± 0.43 | 0.3 ± 0.045 | 2 ± 0.2 |
| 0.1 | 0.05 ± 0.01 | 0.2 ± 0.1 | 0.09 ± 0.02 | 0.7 ± 0.07 |
| 1 | 0.03 ± 0.01 | 0.1 ± 0.06 | 0.03 ± 0.005 | 0.2 ± 0.02 |

A good concordance was noted between the experimental and the predicted repulsive force. When NaCl salt is added in the medium, these two ions come into contact with the dipoles on the opposite side of their charge and create an electric balance. This behavior has the effect of skimming the loads of the double layer and extending the load of dipoles, reducing electrostatic interactions with the sphere and decreasing the length of Debye. As the number of interactions is thus reduced and the length of Debye decreases, the electric field generated by the surface at the height of the sphere also decreases.

*5.3. Repulsive Force on Polymer Film*

Equation (4) also predicts that the repulsive force increases with the density charge of the surface or of the sphere. In order to validate the model, the charge density of the surface was modified by changing the molecule deposited on the surface.

Previous studies demonstrated that polymer film could build up a large repulsion distance to solutes when immersed in an aqueous solution greater than 200 μm [16]. To increase the charge density of the deposited molecule and to better localize it, an electropolymerization of a film was performed. The polypyrole was chosen due to the presence of NH groups and its ability to be deposited locally on electrodes (Figure S7). The electrodeposition was performed by potentiostatic chronoamperommetry on a silicon substrate previously covered by sprayed chrome and gold to enhance its conductivity. Different counter ions (0,1M) were used to determine the impact of each other on the repulsive force. The evolution of the current intensity with time was similar regardless of the supporting salt used (Figure SI-8). However, the charge density values differed from one salt to the other: the highest charge density was obtained with tetrafluoroborate anions when the lowest one was obtained for toluenesulfonate anions, which is consistent with the literature [26]. The PPy/LiClO4 film had a granular structure and covered the whole surface of the substrate (Figure S9) even if it could be seen that the thickness of the film was not entirely uniform due to its surface roughness. The same tendency was observed with the other supporting salts used in this work.

Indeed, the counter-ion used during the synthesis of the polymer film plays an important role in the structure, and thus on the film morphology (Figure S10) [25,26]. Hence, its influence on the repulsive force was studied at pH 2, between a polypyrrole-modified surface and a 10 μm sphere functionalized by APTES (Figure S11) and summarized in Table 3. In Table 3, a repulsive force was measured with an amplitude between 2.5 and 4.9 μN and a repulsive distance upper to 33 μm, which was higher than the previous experiment (Figure 1A (blue curve), and Table 1) with the aminosilane APTES. This increase can be explained by the higher density of the amine groups in polymer films compared to aminosilane-film.

**Table 3.** Repulsive force and distance measured at pH 2 for different counter-ions used during the pyrrole electrodeposition.

| Salt | Interaction Force (μN) | Interaction Distance (μm) |
|---|---|---|
| $LiClO_4$ | $4.9 \pm 0.2$ | $47 \pm 4$ |
| PTS | $2.5 \pm 0.1$ | $33 \pm 3$ |
| $NaNO_3$ | $3.6 \pm 0.1$ | $38 \pm 4$ |
| $NaBF_4$ | $4.1 \pm 0.2$ | $45 \pm 5$ |

Regardless of the counter-ions used, the repulsive forces with the polypyrrole were at least four times higher than with APTES-modified surfaces, which confirms that an increase of the charge density increases the force, as predicted by the model. The repulsive force differences can partially be explained by the morphology of the polypyrrole film, depending on the counter-ion used. Indeed, previous experiments demonstrated that the electrochemical film was only influenced by the anion of the counter-ion regardless of the cation $Li^+$ or $Na^+$ [25,26]. $ClO_4^-$, $BF_4^-$, and $NO_3^-$ anions have a similar size (approximately 6 Å), while PTS anions have a higher diameter (10 Å) [36]. This size difference impacts directly the film morphology: the use of a small anion leads to the formation of a film with a high roughness, whereas the use of a bigger anion as the PTS leads to a more homogeneous and planar structure (Figure S10). We can directly bind the roughness of the substrate at the interaction strengths. Indeed, the higher the roughness of the substrate, the higher the specific surface is. Consequently, the number of interactions increased, and likewise the induced force. Hence, the surface's roughness enables to enlarge electrostatic forces in liquid medium. It should be noticed that for adhesion forces in the air, the impact is the opposite. The roughness reduces the sizes of the contact surfaces induced by local mechanical deformation, and thus reduces the adhesion force [37–39].

## 6. Applications

The repulsive force generated between the surface and the object could be used for micromanipulation tasks. Indeed, if we compare the repulsive force generated by the interaction between a microsphere and a surface in pH 2 (4.9 μN, Table 3) with the weight of the microsphere of borosilicate (11 pN), it appears that it would be easy to place it easily in levitation up to a substrate. This may be interesting for the non-contact micromanipulation to guarantee that a manipulated object will never come into contact with the substrate.

We may also imagine other ways to exploit this high density of charges on micro-objects in non-contact micromanipulation. Indeed, the chemical functionalization generates an important charge density on the sphere, which may induce a high electrophoresis force when located in the electric field. These properties can be used to control the trajectory of the object by electrophoresis. As an example, we consider a sphere functionalized with APTES located in an electric field controlled with an electric voltage located on micro-electrodes. We consider a working space made up of four square electrodes of 100 μm each in aqueous medium, and a voltage applied to each of the electrodes one after the other every 2 s, clockwise. As it is impossible to apply larger tensions when the point of water electrolysis is at 2 V (electrolysis bubbles would perturb object manipulation), we thus consider a voltage of 1.8 V. The results of the simulation presented in Figure 4A show the trajectory of the microsphere. The travel speed increases exponentially between each position (electrode) from 80 μm.s$^{-1}$ at the beginning of the movement to 170 μm.s$^{-1}$ at the end of the trajectory (Figure S12). As the electrophoresis and the drag forces are surface forces, the behavior of the microsphere, and thus its speed, will be the same regardless of the microsphere radius. If we compare these results with the dielectrophoresis, which generates a volume force [5], the manipulation speed varies from 25 μm.s$^{-1}$ to 1000 μm.s$^{-1}$ for an object diameter from 2 μm to 80 μm, respectively. Thus, the electrophoresis combined

with surface functionalization appears to be an interesting alternative to dielectrophoresis when the object size is lower than 10 μm.

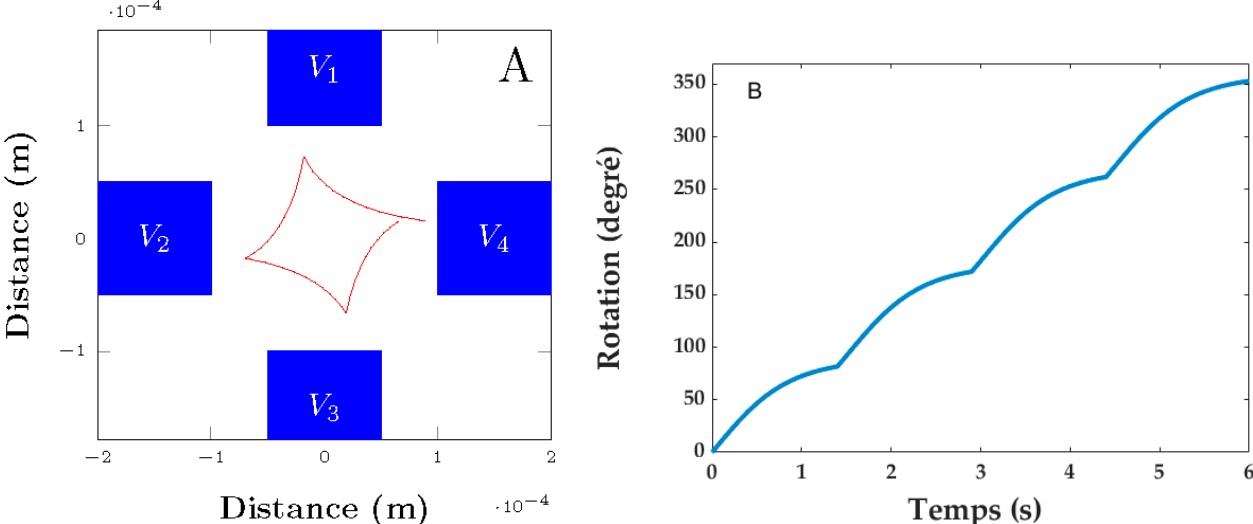

**Figure 4.** (**A**) Trajectory modeling for a microsphere functionalized by APTES by dielectrophoresis; and (**B**) the rotation modeled for a rectangular object (20 μm) functionalized by amine and carboxilic groups, each on a side of the object.

Moreover, several different chemical functions may be placed on different locations on a micro-object to generate more advanced behaviors. Concretely, the orientation of the object may also be controlled by the combination of localized surface functionalization and an electric field. As an example, considering an object having both oxide surfaces and conductive surfaces, a polymer electrodeposition (APTES) can be localized on a conductive surface (Figure S7), whereas the silane molecules could be grafted on oxide surfaces. If the object is grafted with localized amine and carboxilic groups on each side, it generates a high electric dipole (Figure S13). The object will behave as a dipole in an electric field and experiences a high torque which enables it to control its rotation. We demonstrated through simulations that the total rotation of a microsphere may be realized in 6 s with the electrodes defined previously and by applying to the opposite electrodes +0.9 V and −0.9 V simultaneously, every 1.5 s to maximize the speed (Figure 4B).

## 7. Discussion

Controlling the localization of a high number of charges on surfaces may have a lot of applications. This paper illustrates an original way to induce high electrostatic forces on micro-objects along several micrometers and paves the way for the application of non-contact manipulation of micro-objects. The electrostatic force depicted in this article can be used in two different ways in non-contact manipulation.

The first approach consists using the electrostatic force to place the object in levitation several micrometers up to a substrate, and the use of another actuator to move the object parallel to the substrate. In such a case, both the substrate and the micro-object have to be functionalized.

The second approach is closer to electrophoresis, where a functionalized object can be moved in an electric field induced by microelectrodes. The possible performances (typically manipulation velocity) have been compared with more usual principles, such as the dielectrophoresis and our approach, which show a potential interest compared to the state-of-the-art.

Moreover, both approaches (chemical and physical) could be combined in the same device in the future. As an example, the bottom face of the sorting microchannel can have a chemical pattern in order to generate the passive levitation of microparticles in the channel

avoiding sedimentation. The top face could be structured with electrodes enabling to actively sort microparticles.

The long distance of repulsion was already mentioned on several studies with a distance close to 200 μm [32]. Indeed, a large exclusion zone (EZ) was observed in the vicinity of metals [40], hydrogels [41], ion exchange polymer [42], biological tissues [43], white blood cells [29], Nafion polymer [16], and self-assembled monolayer (SAM) [28,44]. An important research was performed to understand such a peculiar phenomenon during the last few years. Several hypotheses were developed as: (i) water structuring [45], (ii) a pH gradient, and thus a charge separation [46], (iii) chemotaxis driven by solute gradients (OH-, H+ and salts in solution) [30], and (iv) a combination of ion-exchange at the surface, diffusion of ions, and diffusiophoresis of particles in the resulting ionic gradients [31]. Recently, a review analyzed a different theory to explain the EZ and concluded several major problems, such as that the water in the EZ undergoes a phase change or significant reordering. They added that Schurr's theory [30] of macroscopic chemotaxis presents a compelling alternative theory, which can explain experimental findings; however, there are still many open questions about exclusion zones.

## 8. Conclusions

In this paper, we have studied the interaction behavior, and more precisely, the repulsive force between a functionalized surface and borosilicate spheres. The experiments were performed as a function of the borosilicate sphere diameter, the medium ionic strength, and the surface charge density. The experimental measurements were compared to a precise numeric model able to predict the interaction forces between a charged surface and a charged micro-object regardless of their shape, and a good agreement was observed. The surface functionalization by polymer electrodeposition allowed the generation of an electrostatic interaction through the electrical charges of chemical origin connected to the ionic strength of the measure medium. This interaction can be characterized by two elements, its strength and its distance. In both cases, the results obtained are innovative and more raised than the repulsive strengths usually met in the chemical systems of typical colloidal suspensions. Because adhesion is the highest current disturbance in micromanipulation (positioning and releasing), the surface functionalization is a promising way to improve micro-robotics' efficiency and accuracy, and to control electrostatic forces in non-contact micro-robotic applications. A wide range of applications in the fields of telecommunications, bioengineering, and more generally speaking, MEMS can also be envisaged for these functionalized micro-grippers.

**Supplementary Materials:** The following supporting information can be downloaded at: https://www.mdpi.com/article/10.3390/app13063886/s1, Equations (S1) to (S-4); Figure S1: Band by band modeling of the sphere; Figure S2: triple electrical layer; Figure S3: Double electrical layer; Figure S4: Modeling of the impact of the sphere band number on the force for a borosilicate sphere with a 10 μm of diameter; Figure S5: Modeling of the repulsive forces and distance between APTES film on surface and on different diameter sizes of a borosilicate sphere glued on the tip extremity; Figure S6: Modeling of the repulsive forces and distance between APTES film on a surface and on a 10 μm borosilicate sphere glued on the tip extremity for different ionic strength; Figure S7: SEM images of the electrodeposition localization of polypyrrole with LiClO4 on a gold electrode; Figure S8: Evolution of charge density with time for different supporting salts; Figure S9: SEM image of the PPy/LiClO$_4$ film; Figure S10: Polypyrrole film morphology versus the counter-ion used: (A) ClO$_4^-$, (B) PTS, (C) NO$_3^-$, and (D) BF$_4^-$; Figure S11: Repulsive force measured at pH 2 at three different points on the polypyrrole film electrodeposited with: A (A) ClO$_4^-$, (B) PTS, (C) NO$_3^-$, and (D) BF$_4^-$; Figure S12: Electrophoresis simulation of functionalized microsphere; Figure S13: Rotation simulation of a micro-object functionalized by two molecules of opposite charges.

**Author Contributions:** All authors contributed to the study. Investigation: A.C. and P.R.; software: P.R., writing—original draft preparation: J.D.; writing—review and editing: M.G., S.L., J.D. and P.R.; supervision: M.G, S.L., and J.D.; project administration and funding acquisition: M.G. All authors have read and agreed to the published version of the manuscript.

**Funding:** This research was partially funded by Franche-Comté Region under FIMICAP (contract "2011C-07333"), by the European project FAB2ASM (contract "FoF-NMP-2010-260079"), by the EIPHI Graduate School (contract ANR-17-EURE-0002)), by the Equipex ROBOTEX project (contract "ANR-10-EQPX-44-01").

**Institutional Review Board Statement:** Not applicable.

**Informed Consent Statement:** Not applicable.

**Data Availability Statement:** Data could be sent upon request to the corresponding authors.

**Acknowledgments:** This work has been supported by the French RENATECH network and its FEMTO-ST technological facility.

**Conflicts of Interest:** The authors declare no conflict of interest.

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
