# Peer review of "Repulsive Force for Micro- and Nano-Non-Contact Manipulation"

_applsci, doi:10.3390/app13063886_

Round 1
Reviewer 1 Report
The presented work “Repulsive force for Micro-& Nano- non-contact manipulation” is quite relevant, interesting and written in good scientific language. The work will certainly be of interest to scientists in the profile field of science. But I suggest Authors should revise the manuscript based on the points below before an acceptance.
· To facilitate the understanding of the work by the readers of the journal, I would recommend to present a graphical abstract of the article, which would reflect the scheme of the experiment.
· Lines 135-140. It is recommended to present the chemical mechanism of surface functionalization of (3-aminopropyl)triethoxysilane, as this may be important for understanding the results of the work (eg lines 158-160 and others).
· Figure 1B shows a significant difference between experiments at 10mM salts (red line). It needs an explanation.
· The results in Table 1 need to be statistically verified, for example using a modified Student's t-test or similar tests. Without such a check, the statement "The experimental data (force values and distance of repulsion) are consistent with 324 the results of the modeling" is not correct. A similar remark is made for Table 2.
· The References contains insufficient profile articles for the last 5 years. It is recommended to update it and update the introduction and discussion of the results.
Reviewer 2 Report
In this paper, a novel approach consists of generating the electric field with electric charges localized via chemical functionalization is presented. The presented method does not require electric connection when using the device. It was shown that electric charges induced by chemical functionalized substrates can generate significant electric field over a long distance. The topic is interesting. However, there are several concerns about this manuscript. The manuscript is poorly written and presented. Applying the following comments and suggestions will improve the current version of the manuscript.
1) The Abstract should be rephrased since it is hard to highlight more important information. Abstracts usually have at least one sentence per each: context and background, motivation, hypothesis, methods, results, conclusions.
2) Arrange the keywords in alphabetic sequence.
3) Author should avoid compounding of the references, e.g [1-3], [7-12], and [18-20] etc.
4) Expansion of the literature review is recommended. In its current form, the presented literature review is limited and several key publications on the topic in hand are overlooked.
5) According to the available technical literature concerning this paper, article innovation is missing in comparison with other research works.
6) The author should provide a critical review about previous studies the investigation of non-contact manipulation.
7) At the end of the introduction, a brief paragraph describing the organization of paper should be added.
8) Section 2.2, the accuracy and validity of the procedures adopted in the "media preparation" should be discussed.
9) It is recommended to describe the mechanism of SAM formation during the salinization.
10) How was the contact angle measured during grafting?
11) Line 130: "protonate the chemical groups of interest.3. Results". "3. Results" should be removed as it seems like a typo.
12) Author should consider improving the formatting and resolution of Figs. 1-4, as some text is hardly readable.
13) Description of Fig. 1 is not clear. It is recommended to use legends in the figure to better understand the presented results.
14) The basic principle used in the current model requires further elaboration.
15) Reasoning should be provided to justify the adopting of classical DLVO theory and neglecting the attractive van der Waals force.
16) The Conclusion section should be expanded and more details regarding the assumptions, limitations and potential future works should be added.
Round 2
Reviewer 2 Report
Authors have improved the manuscript substantially. However, some minor improvements are required still required before the acceptance of the article.
1. Abstract is still a bit confusing.
2. In my opinion, the graphical abstract seems misplaced.
3. The overall structure of the manuscript needs to be consistent.
4. The type of font used in the text within the Figures are different form the one used in the manuscript.
Author Response
- Abstract is still a bit confusing.
We propose a new abstract which better highlight the content of the paper.
- In my opinion, the graphical abstract seems misplaced.
We proposed to the editor to chose between the two graphical abstract. After discussion with him, a new graphical abstract was done.
- The overall structure of the manuscript needs to be consistent.
We modify the location of some paragraph to improve the manuscript.
- The type of font used in the text within the Figures are different form the one used in the manuscript
Type of front in the text and in the figure was now homogeneous.